# Zero-shot Image Editing with Reference Imitation

**Xi Chen**[1]    **Yutong Feng**[2]    **Mengting Chen**[2]    **Yiyang Wang**[1]    **Shilong Zhang**[1]
**Yu Liu**[2]    **Yujun Shen**[3]    **Hengshuang Zhao**[1*]

[1]The University of Hong Kong    [2]Alibaba Group    [3]Ant Group

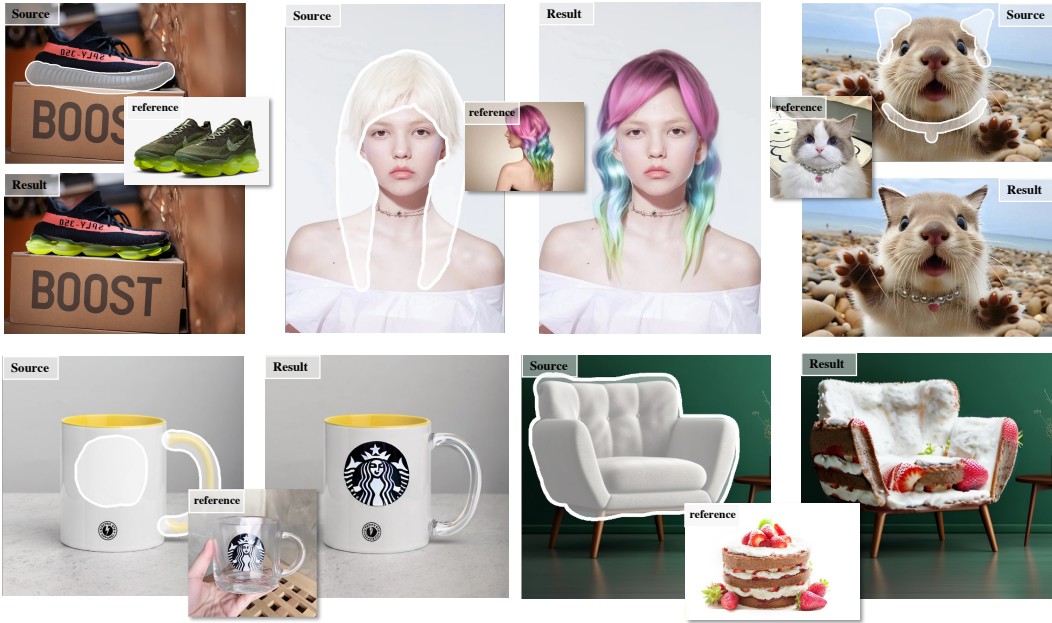

Figure 1: **Diverse editing results** produced by `MimicBrush`, where users only need to specify the to-edit regions in the source image (*i.e.*, **white** masks) and provide an *in-the-wild* reference image illustrating how the regions are expected after editing. Our model automatically captures the semantic correspondence between them, and accomplishes the editing with a feedforward network execution.

## Abstract

Image editing serves as a practical yet challenging task considering the diverse demands from users, where one of the hardest parts is to precisely describe how the edited image should look like. In this work, we present a new form of editing, termed *imitative editing*, to help users exercise their creativity more conveniently. Concretely, to edit an image region of interest, users are free to directly draw inspiration from some in-the-wild references (*e.g.*, some relative pictures come across online), without having to cope with the fit between the reference and the source. Such a design requires the system to automatically figure out what to expect from the reference to perform the editing. For this purpose, we propose a generative training framework, dubbed `MimicBrush`, which randomly selects two frames from a video clip, masks some regions of one frame, and learns to recover the masked regions using the information from the other frame. That way, our model, developed from a diffusion prior, is able to capture the semantic correspondence between separate images in a self-supervised manner. We experimentally show the effectiveness of our method under various test cases as well as its superiority over existing alternatives. We also construct a benchmark to facilitate further research.

---

*Corresponding author.

38th Conference on Neural Information Processing Systems (NeurIPS 2024).

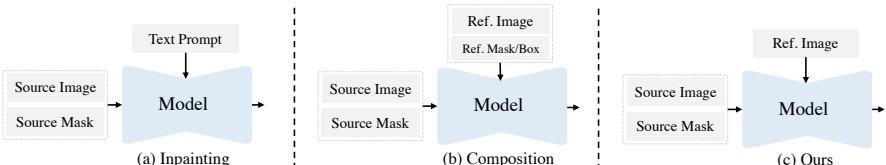

Figure 2: **Conceptual comparisons for different pipelines.** To edit a local region, besides taking the source image and source mask (indicates the to-edit region), inpainting models use text prompts to guide the generation. Image composition methods take a reference image along with a mask/box to crop out the specific reference region. Differently, our pipeline simply takes a reference image, the reference regions are automatically discovered by the model itself.

## 1 Introduction

Image editing enables various applications for creating novel content, *e.g.,* adding new object(s), modifying attributes, or translating image styles. Recently, powered by the large-scale pre-trained text-to-image diffusion models [33, 30, 36], the range of capacity for editing models [18, 3, 7, 2, 43, 15, 26, 12] also expands significantly.

The advanced editing methods could satisfy a large variety of user requirements for modifying either a full image [2, 18] or its local regions [43, 26, 15, 7, 18, 3]. However, it is still challenging for existing editing models to fit the requirements of complicated practical scenarios. For instance, as shown in Fig. 1, it is required to modify the sole of a shoe by referring to another one, or to paste a specified pattern to a given mug. Such kind of editing is important for real applications like product design, character creation, and special effects, *etc*.

For this kind of local editing, existing works take the source image with a binary mask as input. As shown in Fig. 2 (a), inpainting [43, 52] methods re-generate the masked region following text instructions. However, it is not feasible to describe the desired outcomes only with texts. For example, in Fig. 1, the design of shoes or the colors of hair is hard to describe accurately in text. Composition methods [7, 38, 37, 53] take a reference image as input, along with a mask/box representing the reference area, as shown in Fig. 2 (b). They could insert an "individual object" from the reference image into the source image but struggle to deal with local components (like shoe soles and human hair) or local patterns (like logos and texture). These methods require to carefully extract the reference area from the image. Nevertheless, local components are inherently intertwined with the context and are hard to understand when isolated from the whole object. Besides, their [7, 38, 53] training process requires the mask pairs to indicate the same object in different states (*e.g.*, two video frames). Object-level mask pairs are feasible to obtain, but it is difficult to get the part-level pairs at scale.

To deal with the aforementioned requirements, we propose a novel pipeline of editing, termed imitative editing. As illustrated in Fig. 2 (c), given a source image with a masked area for editing, it requires only the reference image without masks. Then, imitative editing targets to fill in the masked area by automatically finding and imitating the corresponding part in the reference image. Such a pipeline formulates more convenient interactions, without strictly separating the reference components from the whole image. Besides, it reaches a harmonious blending referring to the relation between the reference region and its surroundings (*e.g.,* the sole and vape of the shoe).

To achieve imitative editing, we design a framework called `MimicBrush`, with dual diffusion U-Nets to tackle the source and reference images. More specifically, we train it in a self-supervised manner, where we take two frames from a video to simulate the source and reference images. As the video frames contain both semantic correspondence and visual variations, `MimicBrush` learns to discover the reference region automatically and repaint it into the source image with a natural combination to its surroundings. In `MimicBrush`, we send the masked source image into an imitative U-Net and the reference image into a reference U-Net, respectively. Then the attention keys and values of the reference U-Net are injected into the imitative U-Net, which assists in completing the masked regions. As demonstrated in Fig. 1, `MimicBrush` overcomes the variations between the source and reference images in different poses, lightings, and even categories. The generated region highly preserves the details of the visual concepts in the reference image, and harmoniously interacts with the backgrounds. For a more comprehensive evaluation of the proposed method, we also construct a high-quality benchmark of imitative editing. The benchmark includes two main tasks, *i.e.,* part composition and texture transfer. Each task covers several sub-tracks inspired by practical applications, *e.g.,* fashion and product design.

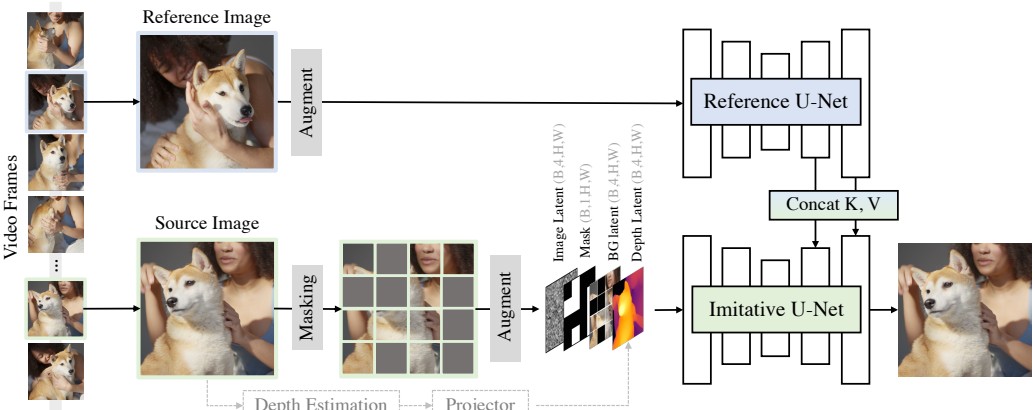

Figure 3: **The training process of** `MimicBrush`. First, we randomly sample two frames from a video sequence as the reference and source image. The source image are then masked and exerted with data augmentation. Afterward, we feed the noisy image latent, mask, background latent, and depth latent of the source image into the imitative U-Net. The reference image is also augmented and sent to the reference U-Net. The dual U-Nets are trained to recover the masked area of source image. The attention keys and values of reference U-Net are concatenated with the imitative U-Net to assist the synthesis of the masked regions.

## 2 Related Work

**Image inpainting.** Traditional image inpainting methods [21, 22, 51, 50] only leverage the background information to complete the masked regions. Powered by the text-to-image diffusion models, recent works [43, 30, 2, 15, 26] leverage text prompts to guide the generation of content of the editing regions. These works fully leverage the flexibility of prompts to generate diversified content. A potential limitation is that only using text prompts could not fully express the user's intention for some specific demands.

**Image customization.** To generate the image for the given subject with high fidelity, customization methods optimize a new "word" or use LoRAs to learn specific concepts. However, most of the customization methods [34, 23, 24, 11, 1, 13] tackles the full object. Besides, they require 3-5 exemplar images and rely on subject-specific fine-tuning which last half an hour. Among them, RealFill [41] and CLiC [35] could customize the local region in context. However, RealFill requires 3-5 images for the same scene, and the finetuned model could only complete local regions of the trained scene. CLiC [35] could customize the local patterns to edit different objects, but it only demonstrates inner-category generalizations and still requires subject-specific fine-tuning.

**Image composition.** This topic explores inserting a given object into a specific location of the background with harmonious blending. The early works involved a long pipeline of image segmentation, pasting, and harmonization [39, 9, 4, 14, 10, 5]. Diffusion-based methods propose end-to-end solutions and support the pose variations of the reference object. Paint-by-example [47] and ObjectStitch [37] use a CLIP image encoder to extract the representation of the object. AnyDoor [7] uses DINOv2 [27] encoders and collects training samples from videos. Later works [53, 54, 28, 44] add camera parameters or text prompts to increase the controllability. However, they mainly focus on inserting the full-object. Part-level composition poses higher demands for modeling the interaction between the editing region and the surrounding context.

## 3 Method

### 3.1 Overall Pipeline

The overall framework of `MimicBrush` is demonstrated in Fig. 3. To realize imitative editing, we design an architecture with dual diffusion models and train it in a self-supervised manner.

Video data contains naturally consistent content, and also shows visual variations such as different postures of the same dog. Thus, we randomly pick two frames from a video clip as the training

samples of `MimicBrush`. One frame serves as the source image, where we mask out some of its regions. Another frame serves as the reference image, assisting the model to recover of the masked source image. Throughout this way, `MimicBrush` learns to locate the corresponding visual information (*e.g.,* the dog's face), and repaint it into the masked area in the source image. To ensure the repainted part is harmoniously blended into the source image, `MimicBrush` also learns to transfer the visual content into the same posture, lighting, and perspective. It is noteworthy that such a training process is built on raw video clips without text or tracking annotations, and can be easily scaled up with abundant videos.

`MimicBrush` leverages a dual branch of U-Nets, *i.e.,* imitative and reference U-Net, taking the source and reference images as input, respectively. The two U-Nets share their keys and values in the attention layers, and are tamed to complete the masked source image by seeking the indications from the reference image. We also exert data augmentation on both images to increase the variation between source and reference image. At the same time, a depth map is extracted from the unmasked source image and then added to the imitative U-Net as an optional condition. In this way, during inference, users could decide whether to enable the depth map of source image to preserve the shape of the objects in the original source image.

## 3.2   Model Structure

Our framework majorly consists of an imitative U-Net, a reference U-Net, and a depth model. In this section, we elaborate on the detailed designs of these components.

**Imitative U-Net.** The imitative U-Net is initialized with a stable diffusion-1.5 [33] inpainting model. It takes a tensor with 13 channels as the input. The image latent (4 channels) takes charge of the diffusion procedure from an initial noise to the output latent code step by step. We also concatenate a binary mask (1 channel) to indicate the generation regions and a background latent (4 channels) of the masked source image. In addition, we project the depth map into a 4-channel depth latent to provide shape information. The original U-Net also takes the CLIP [31] text embedding as input via cross-attention. In this work, we replace it with the CLIP image embedding extract from the reference image. Following previous works [7, 49], we add a trainable projection layer after the image embedding. We do not include this part in Fig. 3 for the simplicity of illustration. During training, all the parameters of the imitative U-Net and the CLIP projection layer are optimizable.

**Reference U-Net.** Recently, a bunch of works [56, 17, 46, 6, 58, 45] prove the effectiveness of leveraging an additional U-Net to extract the fine-grained features from the reference image. For this part, we do not claim novelty and apply a similar design termed reference U-Net. It is initialized from a standard stable diffusion-1.5 [33]. It takes the 4-channel latent of the reference image to extract multi-level features. Following [46], we inject the reference features into the imitative U-Net in the middle and upper stages by concatenating its keys and values with the imitative U-Net as the following equation. In this way, the imitative U-Net could leverage the content from the reference image to complete the masked regions of the source image.

$$\text{Attention} = \text{softmax}(\frac{Q_i \cdot \text{cat}(K_i, K_r)^T}{\sqrt{d_k}}) \cdot \text{cat}(V_i, V_r) \tag{1}$$

**Depth model.** We leverage Depth Anything [48] to predict the depth maps of the unmasked source image as a shape control, which enables `MimicBrush` to conduct texture transfer. We freeze the depth model and add a trainable projector, which projects the predicted depth map (3-channel) to the depth latent (4-channel). During training, we set a probability of 0.5 to drop the input of the depth model as all-zero maps. Thus, the users could take the shape control as an optional choice during inference.

## 3.3   Training Strategy

To fully unleash the cross-image imitation ability of `MimicBrush`, we further propose some strategies to mine more suitable training samples. Considering that our goal is to conduct robust imitative editing even cross categories, the philosophy of collecting training data could be summarized as twofold: First, we should guarantee that the correspondence relation exists between the source and reference images. Second, we expect large variations between the source image and the reference image, which is essential for the robustness of finding the visual correspondence.

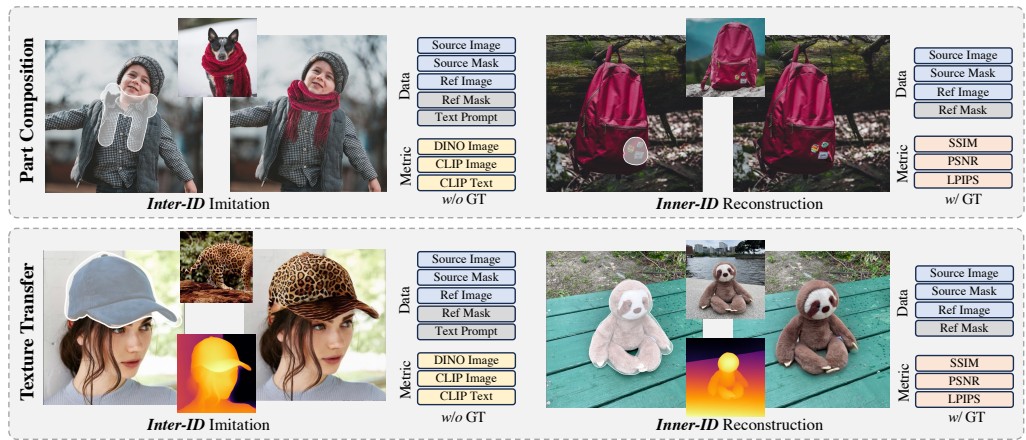

Figure 4: **Sample illustration for our benchmark.** It covers the task of part composition (first row) and texture transfer (second row). Each task includes a *Inter-ID* and *inner-ID* track. The annotated data and evaluation metrics for each track are illustrated beside the exemplar images.

**Data selection.** During training, we sample two frames from the same video. Following [8], we use SSIM [42] as an indicator of the similarity between video frames. We discard the frame pairs with too-big or too-small similarities to guarantee that the selected image pair contains both semantic correspondence and visual variations.

**Data augmentation.** To increase the variation between the source image and the reference image, we exert strong data augmentations. Besides applying the aggressive color jitter, rotation, resizing, and flipping, we also implement random projection transformation to simulate the stronger deformation.

**Masking strategy.** A simple baseline is to divide the source image into $N \times N$ grid and randomly mask each grid. However, we find this purely random masking tends to cause a large portion of easy cases. For example, as the background (*e.g.*, the grassland, the sky) occupies large areas with repeated content/textures, learning to complete these regions does not require the model to seek guidance from the reference image. To find more discriminative regions, we apply SIFT [25] matching between the source and reference images and get a series of matching points. Although the matching results are not perfect, they are sufficient to assist us in constructing better training samples. Specifically, we increase the masking possibility of the grids with matched feature points.

Considering that collecting high-quality images is much easier than videos, we also construct pseudo frames by applying augmentations on the static images and leveraging the object segmentation results for masking the source image. The segmentation masks also improve the robustness of `MimicBrush` to support masks in more arbitrary shapes.

In general, `MimicBrush` does not rely on the heavy annotations of the training data. It fully benefits from the consistency and variation of video data, and also leverages image data to expand the diversity, which makes the training pipeline more scalable.

## 3.4 Evaluation Benchmark

Imitative editing is a novel task, we construct our own benchmark to systematically evaluate the performance. As shown in Fig. 4, we divide the applications into two tasks: part composition and texture transfer, and we set the inter-ID and inner-ID track for each task.

**Part composition** estimates the functions of discovering the semantic correspondence between the source and reference image and compositing the local parts. The inter-ID track aims to composite the local parts from different instances or even different categories. We collect data from the following topics: fashion, animal, product, and scenario. For each topic, we manually collect 30 samples from Pexels [29] thus 120 samples in total, where each sample contain the source and reference image pair. We manually draw the source mask to define the composition requirement. As the generated results do not have ground truth, we annotate the reference regions and write text prompts for the expected result. Thus, we could follow DreamBooth [34] to calculate the DINO [27] and CLIP [31] image similarities between the generated region and the annotated reference region. In addition, we also report the CLIP text similarity between the edited image and the text prompts.

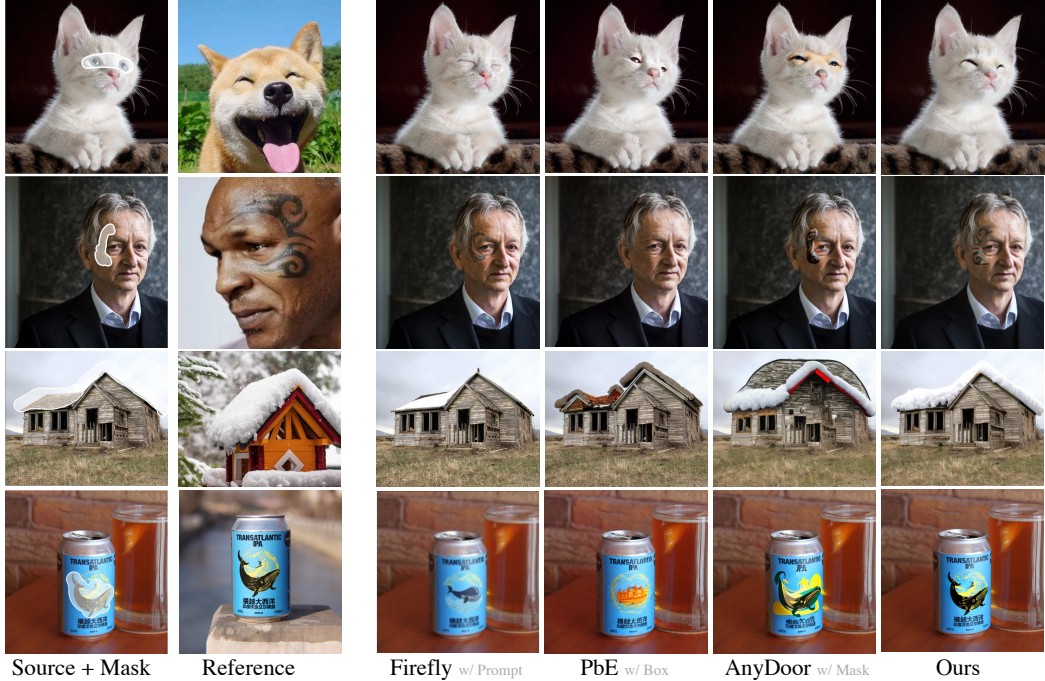

| Source + Mask | Reference | Firefly w/ Prompt | PbE w/ Box | AnyDoor w/ Mask | Ours |

Figure 5: **Qualitative comparisons.** Noticing that other methods require additional inputs. Firefly [32] takes the detailed prompts descriptions. Besides, we mark the specific reference regions with boxes and masks for Paint-by-Example [47] and AnyDoor [7]. Even though, `MimicBrush` still demonstrates prominent advantages for both fidelity and harmony.

We also set an inner-ID track, where we collect 30 image pairs from DreamBooth[34], manually mask the discriminative regions of the source image, and use reference images to complete them. The reference would be an image containing the same instance in different scenarios. Thus, the unmasked source image could serve as the ground truth to compute SSIM [42], PSNR [16], and LPIPS [57].

**Texture transfer** requires strictly maintaining the shape of the source objects and only transferring the texture/pattern of the reference image. For this task, we enable the depth map as an additional condition. Different from the part composition that seeks the semantic correspondence, in this task we mask the full objects thus the model could only discover correspondence between the textures (reference) and the shape (source). We also formulate inter-ID and inner-ID tracks. The former involves 30 samples with large deformations from Pexels [29], like transferring the leopard texture on a cap in Fig. 4. The latter contains an additional 30 examples from the DreamBooth [34] dataset. We follow the same data formats and evaluation metrics as part composition.

## 4 Experiments

### 4.1 Implementation Details

**Hyperparameters.** In this work, all experiments are conducted with the resolution of $512 \times 512$. For the images with different aspect ratios, we first pad the images as a square and then resize them to $512 \times 512$. During training, we use the Adam [19] optimizer and set the learning rate as 1e-5. Experiments are conducted with a total batch size of 64 on $8\times$ A100 GPUs. For the masking strategy of the source image, we randomly choose the grid number $N \times N$ from 3 to 10. We set 75% chances to drop the grid with SIFT-matched features and set 50% chances for other regions. We add the reference U-Net as classifier-free guidance and drop it during training with the probability of 10%. During inference, the guidance scale is 5 as default.

**Training data.** We collect 100 k high-resolution videos from open-sourced websites like Pexels [29]. To further expand the diversity of training samples, we use the SAM [20] dataset that contains 10 million images and 1 billion object masks. We construct pseudo frames by applying strong data augmentations on the static images from SAM and leverage the object segmentation results for masking the source image. During training, the sampling portions of the video and SAM data are 70% versus 30% as default.

Table 1: **Quantitative comparisons for part composition** on our constructed benchmark. The left part of the table reports the evaluation results on the *inner-ID* track. The right part estimates the *inter-ID* track. MimicBrush demonstrates superior performance for each track with the most simplified interaction form. "I" or "T" denotes the image or text similarity.

| | inner-ID | | | inter-ID | | |
|---|---|---|---|---|---|---|
| | SSIM (↑) | PSNR (↑) | LPIPS (↓) | DINO-I (↑) | CLIP-I (↑) | CLIP-T (↑) |
| PbE [47] w/o Box | 0.51 | 15.17 | 0.49 | 41.44 | 81.00 | 29.45 |
| PbE [47] w/ Box | 0.51 | 16.09 | 0.48 | 42.70 | 81.10 | 29.30 |
| AnyDoor [7] w/o Mask | 0.42 | 12.73 | 0.56 | 43.41 | 78.56 | 28.45 |
| AnyDoor [7] w/ Mask | 0.44 | 14.09 | 0.50 | **61.30** | **86.08** | 29.39 |
| MimicBrush | **0.70** | **17.54** | **0.28** | 56.48 | 84.30 | **30.08** |

Table 2: **User study results.** We let annotators rank the results of different methods from the best to the worst from three aspects: fidelity, harmony, and quality. We report both the number of the best picks and the average rank for a comprehensive comparison.

| | Fidelity$_{Best}$ | Fidelity$_{Rank}$ | Hamony$_{Best}$ | Hamony$_{Rank}$ | Quality$_{Best}$ | Quality$_{Rank}$ |
|---|---|---|---|---|---|---|
| PbE [47] w/ Box | 10.8% | 2.64 | 29.2% | 2.57 | 15.8% | 2.59 |
| AnyDoor [7] w/o Mask | 2.8% | 2.46 | 3.0% | 2.35 | 4.2% | 2.45 |
| AnyDoor [7] w/ Mask | 30.6% | 2.77 | 22.6% | 2.72 | 29.4% | 2.71 |
| MimicBrush | **55.8%** | **2.11** | **45.2%** | **2.34** | **50.6%** | **2.23** |

## 4.2 Comparisons with Other Works

In this section, we compare MimicBrush with other methods that could realize similar functions. Noticing that imitative editing is a novel task, no existing methods could perfectly align our input formats. Thus, we allow additional inputs for other methods. For example, we give additional masks or boxes for AnyDoor [7] and Paint-by-Example [47] to indicate the reference regions. We also pick the state-of-the-art inpainting tool Firefly [32] and feed it with detailed text descriptions.

**Qualitative comparison.** We visualize the qualitative results in Fig. 5. Although Firefly [32] accurately follows the instructions and generates high-quality results, it is hard for the text prompt to capture the details of the desired outputs, especially for logos or patterns like tattoos. Paint-by-example [47] requires a cropped reference image in which the reference regions are centered. However, even if we provide this kind of input, as this model only uses a single token to represent the reference, it cannot guarantee the fidelity between the generated region and the reference region

We carefully annotate the masks of the reference region and fed them to AnyDoor [7]. It demonstrates stronger abilities for identity preservation but fails to synthesize harmonious blending. We analyze that there are two main reasons: first, some local parts could not be well understood when cropped out of the context. Second, most of the training samples of AnyDoor [7] are full objects. It requires paired mask annotation for the same instance in different video frames for training. The paired masks are feasible to collect for full objects but hardly feasible for local parts. However, MimicBrush gets around this problem by leveraging the model to learn the correspondence itself in the full context instead of using the paired masks. In this way, MimicBrush shows significant superiorities compared with previous methods for completing arbitrary parts with a full reference image.

**Quantitative comparison.** We also report the results on our constructed benchmark for part composition in Tab. 1. For the inner-ID track with ground truth, MimicBrush shows the dominant performance even though we give additional conditions for other works. For inter-ID imitation, it is more challenging to discover the correspondent reference region. MimicBrush could still get competitive performance compared with AnyDoor [7]. We should note that the reference mask is given to AnyDoor. Therefore, it could forcedly locate the reference regions thus taking some advantages for the evaluation. However, as demonstrated in Fig. 5, it struggles to generate harmonious blending and preserve the fine details.

**User study.** Considering the metrics could not fully reflect human preferences to some extent, we also organized a user study. We let 10 annotators rank the generation results of different models on our benchmark (introduced in Sec. 3.4). We evaluate each sample from three aspects: fidelity, harmony, and quality. Fidelity considers the ability to preserve the identity of the reference region. Harmony estimates whether the generated regions could naturally blend with the background. Quality regards whether the generated regions contain fine details in good quality. Those three aspects are evaluated independently. Results are listed in Tab. 2, MimicBrush earns significantly more preferences.

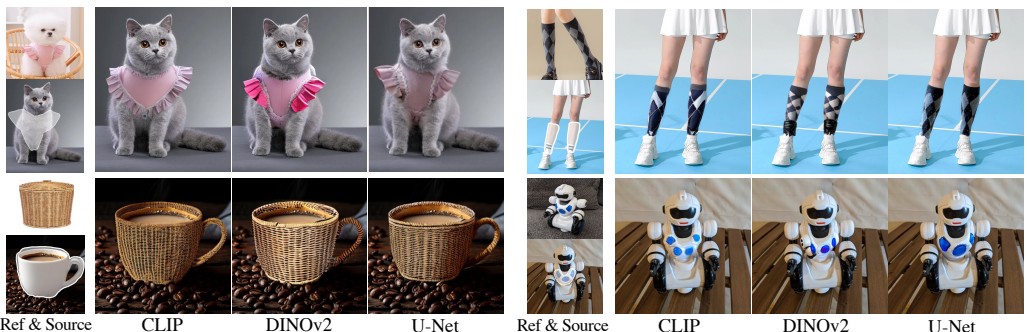

Figure 6: **Ablation study for reference feature extractors.** CLIP and DINOv2 encoders could also achieve imitative editing but lag behind the U-Net in preserving the fine details.

Table 3: **Ablation study for different reference feature extractors**. U-Net demonstrates consistent advantages across different evaluation tracks and metrics compared with CLIP and DINOv2.

| | Part Composition | | | | | | Texture Transfer | | | | | |
|---|---|---|---|---|---|---|---|---|---|---|---|---|
| | SSIM (↑) | PSNR (↑) | LPIPS (↓) | DINO-I (↑) | CLIP-I (↑) | CLIP-T (↑) | SSIM (↑) | PSNR (↑) | LPIPS (↓) | DINO-I (↑) | CLIP-I (↑) | CLIP-T (↑) |
| CLIP Encoder | 0.66 | 16.78 | 0.31 | 45.03 | 82.3 | 30.05 | **0.75** | 16.78 | 0.31 | 37.86 | 78.30 | **31.39** |
| DINOv2 Encoder | 0.67 | 16.50 | 0.32 | 48.34 | 82.40 | 29.84 | 0.74 | 17.27 | 0.27 | 46.34 | 78.00 | 30.61 |
| Ours (U-Net) | **0.70** | **17.54** | **0.28** | **56.48** | **84.30** | 30.08 | **0.75** | **17.73** | **0.26** | **49.83** | **79.44** | 30.75 |

Table 4: **Ablation study for training strategies**. In the first block, we verify the importance of training data and augmentation. In the second block, we explore different strategies for masking the source image. The performance of our full pipeline is given at the bottom.

| | Part Composition | | | | | | Texture Transfer | | | | | |
|---|---|---|---|---|---|---|---|---|---|---|---|---|
| | SSIM (↑) | PSNR (↑) | LPIPS (↓) | DINO-I (↑) | CLIP-I (↑) | CLIP-T (↑) | SSIM (↑) | PSNR (↑) | LPIPS (↓) | DINO-I (↑) | CLIP-I (↑) | CLIP-T (↑) |
| ◦ Image Data Only | 0.67 | 14.95 | 0.33 | 39.68 | 79.90 | 29.12 | 0.70 | 15.10 | 0.31 | 41.30 | 77.80 | 30.72 |
| ◦ Weak Aug. | 0.68 | 16.98 | 0.30 | 50.55 | 83.20 | 29.81 | 0.74 | **18.13** | **0.26** | 50.92 | 80.0 | 31.23 |
| ◦ Single Box$_{0.50}$ | 0.66 | 15.97 | 0.32 | 47.41 | 82.10 | 28.97 | 0.72 | 16.24 | 0.31 | 48.52 | 78.10 | 29.30 |
| ◦ Mask Gid$_{0.25}$ | 0.68 | 17.10 | 0.30 | 49.17 | 82.94 | 29.80 | 0.74 | 17.35 | 0.27 | 50.09 | 79.50 | 31.05 |
| ◦ Mask Gid$_{0.50}$ | 0.68 | 16.97 | 0.30 | 50.09 | 82.56 | 29.84 | 0.73 | 17.25 | 0.27 | 48.58 | 81.00 | 30.35 |
| ◦ Mask Gid$_{0.75}$ | 0.67 | 16.61 | 0.30 | 49.94 | 83.00 | 29.75 | 0.73 | 16.69 | 0.28 | **52.46** | **81.75** | 30.02 |
| ● MimicBrush | **0.70** | **17.54** | **0.28** | **56.48** | **84.30** | **30.08** | **0.75** | 17.73 | **0.26** | 49.83 | 79.44 | **31.75** |

## 4.3 Ablation Studies

In this section, we conduct extensive ablations to verify the effectiveness of different components.
**Referece feature extractor.** `MimicBrush` leverages a dual U-Net structure to model extractor the features from the source and reference image respectively. Some previous works [40, 55] prove that the pre-trained diffusion models contain strong prior to capture semantic correspondence. We explore whether an asymmetric structure could still learn the semantic correspondence under our self-supervised training pipeline. We replace the reference U-Net with the CLIP/DINOv2 image encoder and inject the $16 \times 16$ patch tokens within the cross-attention layers. Visual comparisons are provided in Fig. 6, CLIP and DINOv2 also successfully locate the reference region, but U-Net illustrates clear superiorities for preserving the fine details. We also conduct quantitative results in Tab. 3, where we conclude that CLIP and DINOv2 also reach competitive performance. However, as U-Net gives multi-level representations with higher resolutions, and the feature space is naturally aligned with the initiative U-Net, it gives better results when serving as a reference feature extractor.

**Training strategies.** In Tab. 4, We first verify the effectiveness of the "video-based" training pipeline. When using the statistics images only, the performance for each task drops significantly. It shows that object deformation or variation in videos is vital for realizing imitative editing. Afterward, we remove the strong color jitter, resizing, and projection transformation. We observe a clear degradation in part composition, specifically for the inter-ID track. It verifies the importance of augmentation for robust semantic correspondence matching.

In the second block, we explore the different masking strategies for the source image and report the performance using a single box and different gid ratios. As introduced in Sec. 3.3, these purely random masking strategies could cause a large number of low-quality training samples. In contrast, we leverage SIFT matching to enhance the masking and reach better performances (bottom row).

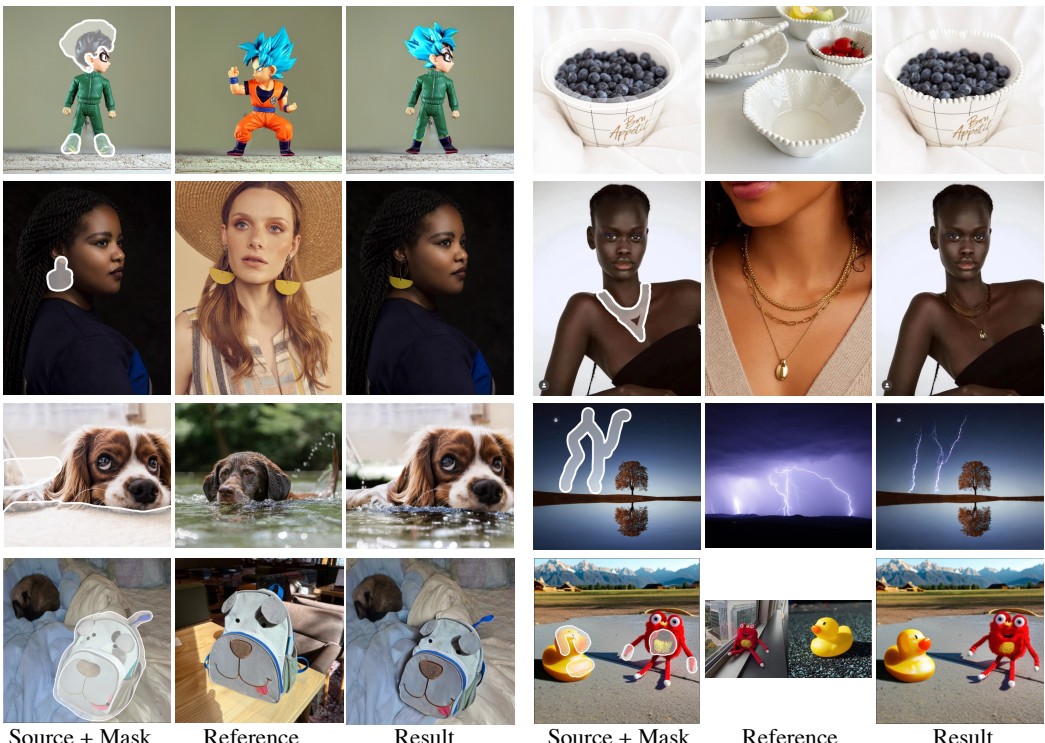

| Source + Mask | Reference | Result | Source + Mask | Reference | Result |

Figure 7: **Diverse applications** supported by `MimicBrush`. Our methods could be applied conveniently for product design, accessories wearing, editing the scene images, and refining the flawed generation results of other methods. `MimicBrush` is able to edit multiple regions in one pass.

## 4.4 Qualitative Analysis

In this section, we give more visual examples and discuss the potential applications. As demonstrated in Fig. 7, `MimicBrush` could deal with images from various topics and domains. The first row illustrates the application for product design. The second row shows some examples of jewelry-wearing. It should be noticed that the segmentation masks for the necklace are hard to extract, but `MimicBrush` gets rid of the segmentation step and directly transfers the necklace from the reference image to the source image. In the third row, we show that `MimicBrush` could also deal with backgrounds also nature effects, proving its strong generalization ability.

The last row illustrates a practical application that we could leverage `MimicBrush` as a post-processing to refine the generation results of other works. In the left example, we improve the fidelity for the image generated by AnyDoor [7]. In the right example, we mark multiple to-edit regions in the source image generated by Cones-2 [24] and provide a concatenated reference image containing both objects. We observe that `MimicBrush` could refine all the to-edit regions in a single pass.

## 5 Conclusion

We present a novel form of image editing with simple interactions, called imitative editing. In our setting, users are only required to mark the editing region on the source image and provide a reference image that contains the desired visual elements. `MimicBrush` automatically finds the corresponding reference region to complete the source image. To achieve imitative editing, we take full advantage of the consistency and variation of videos and design a self-supervised training pipeline that uses one frame to complete the masked regions of another frame. `MimicBrush` demonstrates impressive performances for various editing tasks and supports a wide range of applications. To facilitate future explorations, we construct a benchmark to comprehensively evaluate imitative editing. This work is expected to bring new inspiration for the community to explore more advanced techniques for image generation and editing.

**Limitations & potential impacts.** `MimicBrush` demonstrates robust performance. However, it could fail to locate the reference region when the region is too small or multiple candidates exist in the reference image. In this case, users should crop the reference image to zoom in on the desired regions. `MimicBrush` could deal with a wide range of images, thus making it possible to produce some content with negative impacts. Therefore, we would add censors to filter out the harmful content when we release the code and demo.

**Acknowledgement.** This work is supported by the National Natural Science Foundation of China (No.62201484), HKU Startup Fund, and HKU Seed Fund for Basic Research.

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

# Appendix

In the Appendix, we begin by providing further descriptions of our constructed benchmark in Appendix A, along with the proposed metrics. Subsequently, we delve into the details of each step of the user study in Appendix B. Following that, we offer additional analysis and discussion on the training strategies in Appendix C. Finally, in Appendix D, we include more visual demonstrations and qualitative analysis for various applications.

## A  Benchmark Details

In this section, we provide detailed descriptions of our constructed benchmark and the proposed metrics. We include examples of part composition and texture transfer, and we visualize the annotation for each track in Fig. A1. For the inter-ID tracks, we present a source image, source mask, reference image, reference mask, and a text prompt to specify the expected results. The reference mask and text prompts are used solely for evaluation, where we calculate DINO and CLIP image similarities, as well as CLIP text similarity. Specifically, to compute the DINO and CLIP image similarities, we first remove the background of both the reference image and the generated results using the reference masks and source masks. Then, we calculate a minimum bounding box for the two masks, cropping both the generated and reference regions accordingly. The DINO and CLIP similarities are calculated between these two cropped regions with clean backgrounds.

For inner-ID tracks, since the source image can be considered as the ground truth, we omit annotating prompts for CLIP test scores. Instead, we compute the SSIM, PSNR, and LPIPS metrics. Similar to the cropping operation used before calculating the DINO and CLIP image similarities, we also utilize the masks to obtain the bounding box around the generated region. This box is then employed to crop the corresponding regions from both the generated result and the source image for computing the SSIM, PSNR, and LPIPS. Although the reference masks are not utilized for model input or metric computation, we still annotate the masks for each sample. This enables evaluation for other methods like Paint-by-Example [47] and AnyDoor [7].

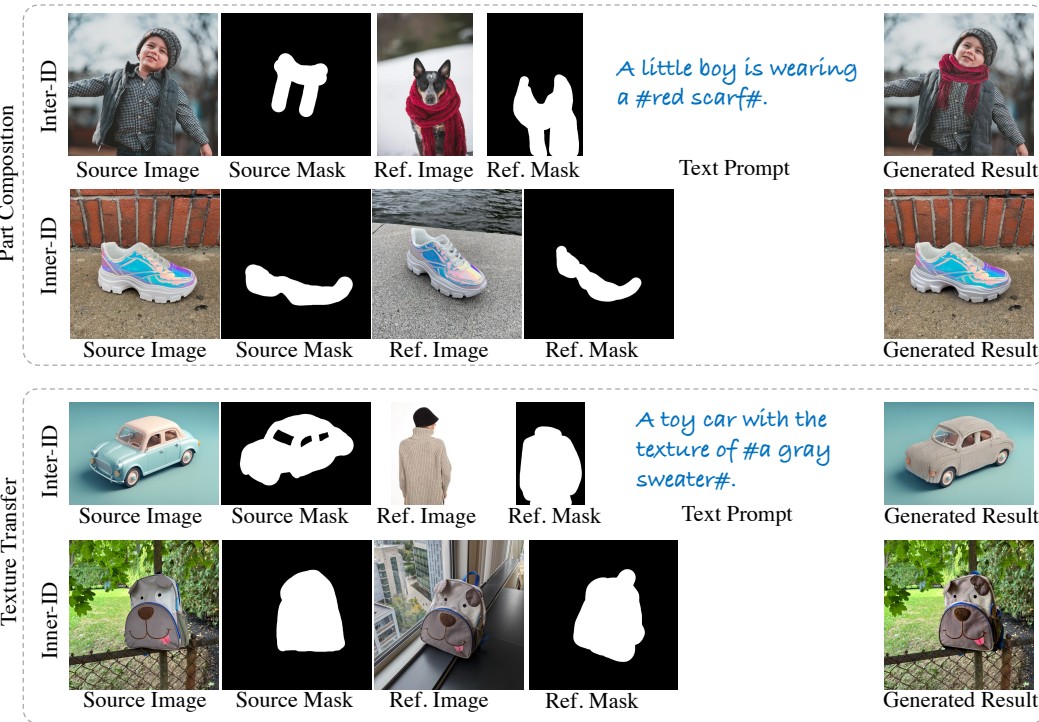

Figure A1: **Visual demonstrations** for the samples in our constructed benchmark. We provide one example for each track to illustrate the detailed data formats.

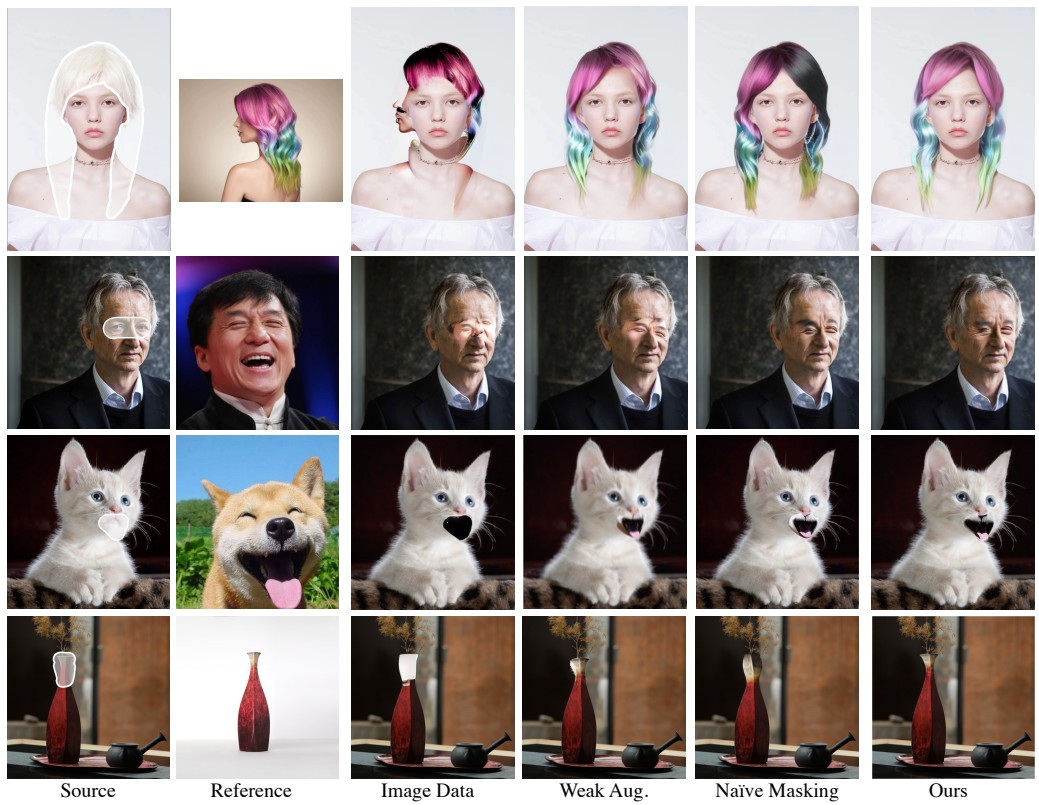

| Source | Reference | Image Data | Weak Aug. | Naïve Masking | Ours |

Figure A2: **Qualitative ablations** for different training strategies. We present the results for various training strategies, including training with only image data, applying weak augmentations, and utilizing naive masking strategies. For comparison, we also provide the generation results of our full-version model.

## B  User Study Details

As outlined in the main paper, we conducted a user study involving 10 annotators to assess the generation results of various methods. We devised a user interface that presents the source image, source mask, reference image, and reference mask on the left side to clarify the expected editing outcomes. On the right side, the user interface displays the four results predicted by four different methods, each marked with an index. Annotators were tasked with answering four single-choice questions regarding the best, second-best, third-best, and fourth-best results index. To enhance the reliability of the user study results, we implemented several measures:

First, we provided comprehensive documentation and rating examples for each task. Annotators were initially tasked with rating a small subset of data. Once their annotations passed our examination, they continued with the remaining samples.

Second, to prevent annotators from distinguishing the results of different methods based on additional information, we standardized the image size of each result and shuffled the order of the four results randomly for each sample. We recorded the shuffled indexes to maintain consistency.

Third, as human evaluation was required from three different aspects (fidelity, harmony, quality), these aspects were evaluated as three separate annotation tasks to ensure independence and accuracy.

## C  Training Strategy Analysis

In this section, we provide additional analysis and qualitative ablation results for the training strategies. We follow the experimental settings introduced in Tab. 4 of the main paper. As depicted in Fig. A2, when only utilizing statistical images for training, the first two rows indicate that the model is still

capable of learning semantic correspondence. However, it struggles to adapt to pose and lighting variations, resulting in rigid blending. This underscores the importance of incorporating visual variations from video data for our method to conduct robust and generalized imitative editing. We also showcase generation results using weak data augmentation and a naive masking strategy (*i.e.*, masking the random grid with 50% probability). Both approaches exhibit clear performance degradation compared to our full version of `MimicBrush`, which employs strong data augmentation and a matching-guided masking strategy.

## D   More Visualization Results

This section presents a comprehensive array of examples highlighting the exceptional capabilities of our methodology in diverse contexts.

Fig. A3 showcases results from inner-ID reconstruction where a specified region on the initial image is repainted using another image of the same instance as a reference. These outcomes illustrate our method's precision in accurately identifying the target region and retaining intricate details with high fidelity.

In Fig. A4, further examples of part composition are provided. The first row demonstrates a seamless editing process, which facilitates the combination of multiple visual elements in a cohesive manner. The next two rows exhibit variations across assorted topics and domains, showcasing our method's robust adaptability.

In Fig. A5, we present instances of texture transfer applications. Here, we utilize the depth map to maintain the original structure of the source image while artfully transferring texture from the reference image. This technique facilitates a myriad of imaginative designs, allowing creators to infuse elements with new aesthetic qualities while respecting the underlying form. Such versatility not only provides a tool for artists to expand their creative palette but also offers practical solutions for industries requiring rapid visualization of texture variations without altering the product's shape. Consequently, our method stands as a testament to the convergence of creativity and technical innovation, opening new avenues for design exploration.

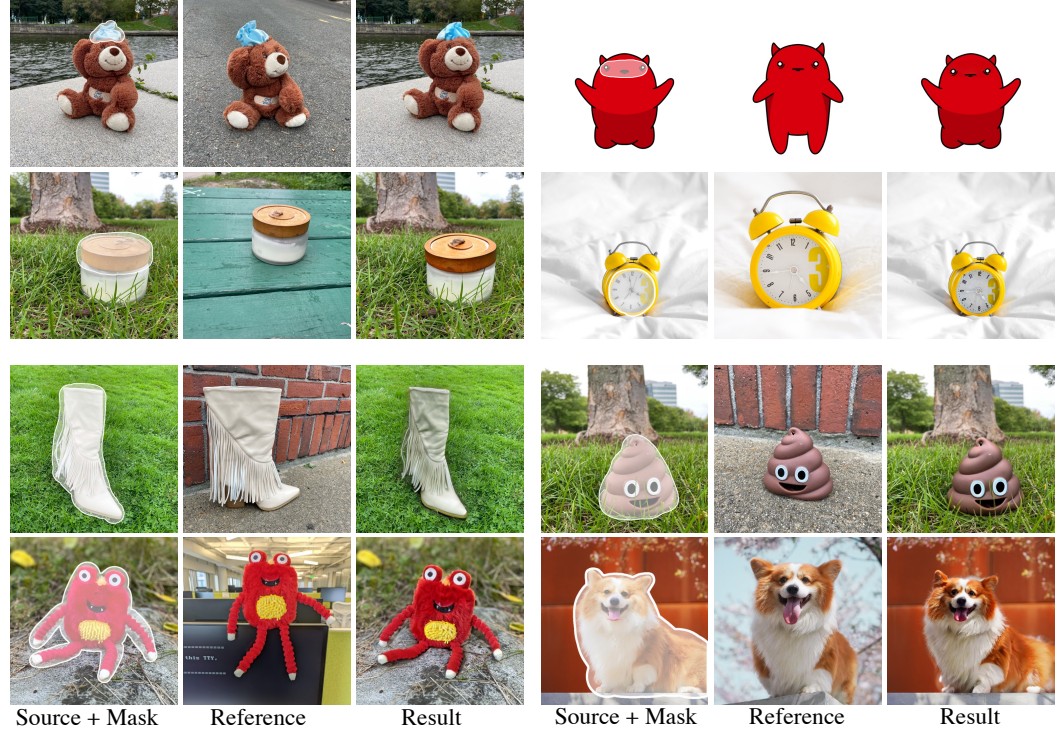

Source + Mask    Reference    Result    Source + Mask    Reference    Result

Figure A3: **Demonstrations for inner-ID reconstruction.** In this scenario, we repaint a masked local area of the source image using a reference from the same instance. For the final two rows, we implement shape control utilizing the depth map for enhanced precision.

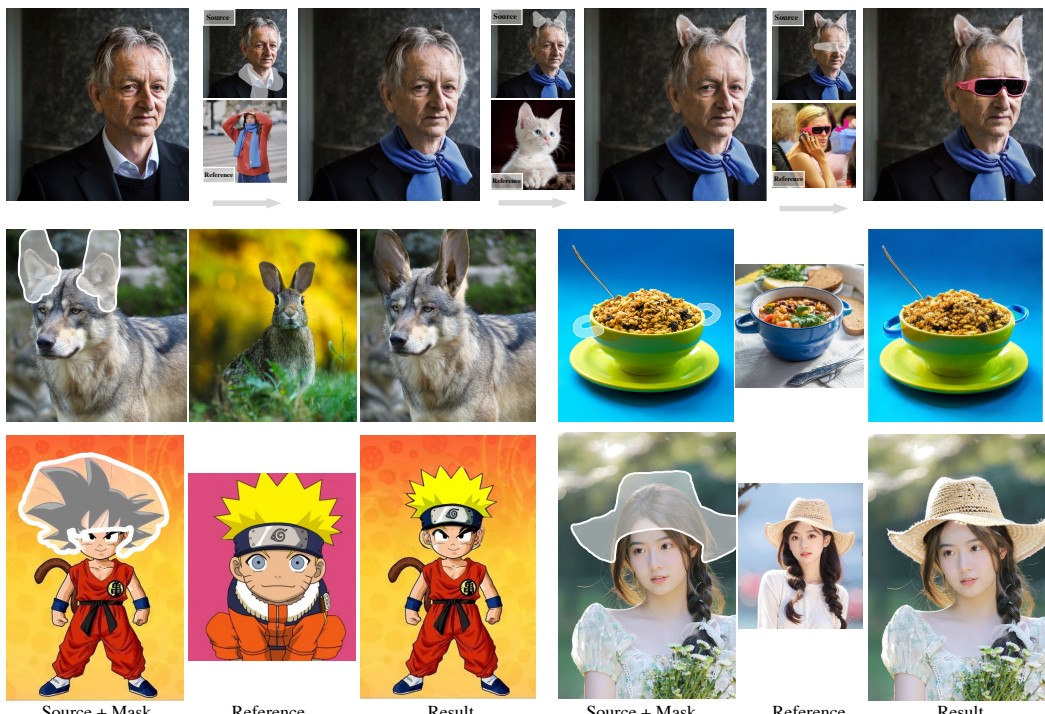

Source + Mask    Reference    Result    Source + Mask    Reference    Result

Figure A4: **Demonstrations for part composition.** The first row illustrates a case of continuous editing, allowing for a step-by-step composition of multiple visual elements. The subsequent rows provide examples across various themes such as animals, fashion, and product design.

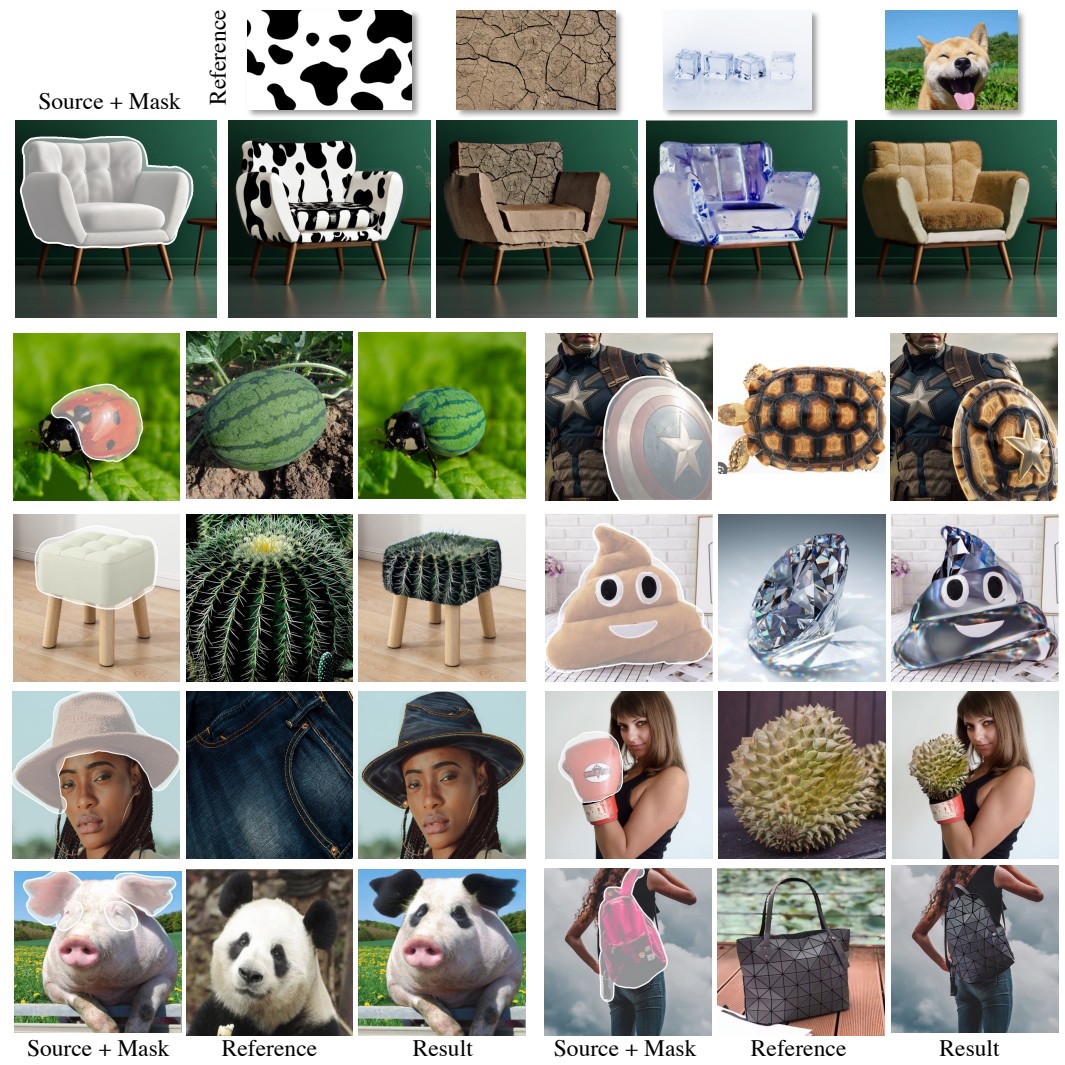

Figure A5: **Demonstrations for texture transfer.** When applying the shape control, `MimicBrush` could strictly follow the original shape and transfers the novel texture from the reference images.

