# OpenReview forum: "Zero-shot Image Editing with Reference Imitation"
_NeurIPS.cc/2024/Conference — NeurIPS 2024 poster_

### Official Review · Reviewer_q758 · 2024-07-01

**Soundness:** 3
**Presentation:** 3
**Contribution:** 2
**Rating:** 5
**Confidence:** 4

**Summary:**

The paper proposes a method to modify specific parts of the image with the content from the reference image and consistently with the original content. It combines the technique of dual diffusion to replace the key and value features, which is used in MasaCtrl, of the source image with the reference image. The method is trained in a self-supervised manner, in which some parts of the image are masked, and the model is trained to recover them.

**Strengths:**

- The paper is well-written and easy to understand.
- The end-to-end pipeline doesn’t need fine-tuning for each image.
- The proposed method is friendly for users that only give the source image, reference image, and mask, and then the model automatically gets the correct content from the reference image and aligns for the original part that is specific by mask.
- Experimental results are impressive with various categories and metrics.

**Weaknesses:**

- The proposed method is not actually zero-shot as it requires training. It requires expensive resources for the training model.
- The method presented in this paper lacks novelty. It uses the old technique (i.e., replacing the key, value feature from MasaCtrl [MasaCtrl: Tuning-Free Mutual Self-Attention Control for Consistent Image Synthesis and Editing, ICCV 2023]) and combines with an old training strategy that recovers the region that is specific by mask.
- The source and reference images must be the same scale as the object, and the source image must contain only one or a salient object.

**Questions:**

What if the source image is a complicated image that contains some objects and the desired modified object is not a salient object?

**Limitations:**

The paper is not so novel; it reuses components from existing techniques. The proposed method is not actually zero-shot as it requires training.

---

> ### Author Rebuttal · Authors · 2024-08-07
>
> ``W1. The proposed method is not actually zero-shot as it requires training. It requires expensive resources for the training model.``
>
> **This comment is erroneous**. “Zero-shot” is different from “training-free.” Zero-shot means that we train the model on some examples, and the model’s ability can generalize to unseen examples without case-by-case fine-tuning or optimization. Hence, our setting is exactly "zero-shot."
>
> Additionally, our model is not training-expensive. Our final model is trained on 8 GPUs, and we have verified that our results can be reproduced with 2 or 4 GPUs. Besides, we do not require any additional human annotations for training our model.
>
>
> ``W2. The novelty and techniques of our methods.``
>
> We disagree. There exist misunderstandings about our motivation and novelty.
>
> First, the dual u-net structure similar to MasaCtrl is not our contribution, it is a common practice for injecting the features of reference images, which is widely used in recent papers [56, 17, 46, 6, 58, 45]. Besides, we tackle totally different tasks with MasaCtrl.
>
> Second, the core training strategy is not the MAE-like mask completion, but "correspondence learning" between two images.  We fully utilize the consistency and variation of video frames, randomly mask one video frame, and use another "full video frame" for completion.
>
> Please refer to **A. Core contributions and novelty.** in the global rebuttal block to see our contribution and novelty.
>
>
> ``W3. The source and reference images must be the same scale as the object, and the source image must contain only one or a salient object.``
>
> It is not true.
>
> First, we support source and reference objects of different scales. For example, in Fig. 5, the second row features Hinton and Tyson, and in Fig. A4 in the appendix, the first and second rows illustrate the scarf and ears. These examples all have quite different scales Additionally, as mentioned in L145-147, we add strong augmentations like resizing during training, making our model robust to scale variance.
>
> Second, the source image can contain multiple objects. As shown in the bottom-right example of Fig. 7, our model performs well in one pass. In this paper, we primarily illustrate images with a single salient object for clear visualization. However, our method can support complex source images, and we have added more examples in Fig. R1 in the rebuttal PDF, where the source images contain **complicated scenes and large scale-variations**. We will include these examples in the revision.

---

> ### Author Response · Authors · 2024-08-11
>
> Dear Reviewer,
>
> We kindly remind you about our submitted rebuttal. We understand that your time is valuable, and we greatly appreciate the effort you put into reviewing our work. If you have any questions or require further clarification on any points, please don't hesitate to reach out. We look forward to your feedback and hope for a favorable consideration.

---

> ### Author Response · Authors · 2024-08-13
>
> Dear Reviewer q758,
>
> As the deadline for the discussion phase approaches, we wanted to kindly remind you to read our rebuttal. We highly value your feedback and are eager to incorporate your suggestions to improve our work.
>
> If there are any aspects that require further clarification or additional information, we would be more than happy to engage in continued discussion and provide any necessary details.

---

> ### Comment · Reviewer_q758 · 2024-08-13
>
> Dear Authors,
>
> Thank you for your detailed rebuttal.
>
> First, I would like to acknowledge that I agree with your definition of the zero-shot method.
>
> Second, while I appreciate the detailed explanation of your contributions and their novelty, I believe that the novelty is primarily focused on user experiments related to image editing tasks. From your presentation in the paper, it seems that you are proposing a novel architecture (Imitative U-Net). However, I still maintain that the overall method may not be entirely novel.
>
> Finally, the examples provided in Fig. R1 do not fully convince me. Could you test a more complex example that includes editing smaller objects within the image? For instance, an image with a large number of overlapping objects or objects that do not interact with each other would be more compelling.
>
> In conclusion, I would like to keep my rating as 'BorderLine Reject.’

---

> ### Author Response · Authors · 2024-08-13
>
> Thank you for your response. I believe that the contribution and novelty are still misunderstood.
>
> `` it seems that you are proposing a novel architecture (Imitative U-Net)``
>
> We claim multiple times that **the dual U-Net design is not our novelty/contribution, instead, it is a common practice**. We do not claim contribution. As mentioned in line 121, this structure is widely used in many tasks that require a reference image [56, 17, 46, 6, 58, 45]. These works verified the advantages of "dual U-Net" over previous works like IP-adapter or controlnet. We leverage this type of design solely as our feature injection module and do not claim it as our novelty.
>
> `` However, I still maintain that the overall method may not be entirely novel. ``
>
> We are the **only work** that could realize local region editing with a reference image ***even without providing the reference mask***, for this motivation, please refer to "B. Motivation of getting rid of the reference mask" in the global rebuttal.
> In addition, we are the **only work** that trains the model using one frame to complete another frame to learn the correspondence. As **the whole task and training strategy** are completely novel, we believe the novelty is enough. Again, we would like to reaffirm our core novelty below.
>
> **Novel form for image editing.** Unlike the commonly studied task of "object insertion", our proposed "imitative editing" offers a new editing form, which does not require users to specify the region of interest from the reference image. Such a design sufficiently simplifies the editing process, because users only need to brush once (i.e., the source image) instead of twice (i.e., both the source image and the reference image), and hence clearly distinguishes our approach from existing methods. It is also noteworthy that getting rid of the reference mask is essential for part-level editing, making our algorithm more flexible in practice.
>
> **Novel training strategy.** To accomplish "imitative editing", it requires the model to automatically analyze the semantic correspondence between the to-edit region of the source image and the "full" reference image. For such a purpose, our approach benefits more from the training strategy but not the data. Concretely, we propose to leverage the temporal consistency and variation across video frames, and urge the model to recover the masked region of one video frame by only using the information from another frame without specifying the corresponding region. To our knowledge, our proposed training pipeline offers a brand new way of using video data for diffusion-based editing.
>
> **Benchmark for the novel task**. We construct a diversified benchmark that contains multiple tracks and covers a large variety of scenarios. Besides, we come up with various metrics to thoroughly evaluate the compare different methods. Our benchmark could be beneficial for further explorations for imitative editing.
>
> ``Could you test a more complex example that includes editing smaller objects within the image?``
>
> Please notice that, in the second row of Fig.R1 of the rebuttal PDF, we have added examples for editing "the candle", "the hat", and even "the socks of a full-body human".  **We have already shown these small things that occupy less than 5% pixels of the source images**.
>
> We kindly remind you that, as the "to-edit" region could be specified by users with a "source mask". In this way, **there would not be any challenges** in distinguishing the "to-edit" region even if the environment is complicated with distracters or this region is small.
>
> According to the rebuttal policy, the PDF files could not be updated. We would contact ACs to see if we could give you more examples.
>
>
> **Please check if this response addresses your concerns, if you have further questions, we are happy to discuss them.**

---

> ### Author Response · Authors · 2024-08-14
>
> Dear Reviewer,
>
> We have added examples for even smaller objects and given the link to ACs, you may receive this link from ACs later. Here we describe the examples:
>
> - A camera shot filled with large numbers of fruits, where some fruits are partially obscured by others. We edited one of the fruits that was partially obscured.
>
> - A complex rural scene with more than ten people and ten dogs or sheep, where there is significant occlusion between the people and animals. We replaced one of the heavily obscured dogs with a tiger.
>
> - A windowsill filled with plants (more than 10), with some pots partially obscuring others. We edited one of the pots according to the provided reference image.
>
> Please note that editing very small areas is only a corner case with limited practical usage in real-world applications and is unrelated to the main focus of our paper. However, we have still provided the requested samples. We hope these examples fully address your concerns. If you have any new questions, feel free to reach out to us at any time.

---

> ### Comment · Reviewer_q758 · 2024-08-14
>
> Dear Authors,
>
> Thank you for your further explanations.
> My concerns are mostly addressed, but it is still not so convincing.
> I can only increase my rating to Borderline Accept.
>
> I already know that the dual U-Net design is not your novelty/contribution. One thing I want to point out is that the way your contribution is presented in the paper confused me. I don't think your method is actually novel; it is only novel in the context of user experiments.
>
> p/s: I haven't received your examples of small objects.

---

> > ### Author Response · Authors · 2024-08-14
> >
> > Dear Reviewer,
> >
> > Thank you for your acknowledgment.
> >
> > ``the way your contribution is presented in the paper confused me``
> >
> > Thank you for pointing it out. To make our contribution clearer, we would add a small paragraph at the end of the introduction section to summarize our contributions and novelties. In addition, we would add an explanation paragraph in Section 3.2 to claim the dual U-Net is not the contribution of this paper in the revised version.
> >
> > `` It is only novel in the context of user experiments.``
> >
> > First, image editing is an application-oriented task, and we formulate a more convenient editing form without the reference mask. We think "this novelty in user experience" is already quite important, and could make great contributions for the image editing community.
> >
> > Besides, this "novelty for users" could not be simply archived by applying existing techniques, it also requires the support of our novel training strategies.  Our video-based training strategy is specifically designed for learning semantic correspondence without reference masks, which is important for realizing our "imitative editing".
> >
> > ``p/s: I haven't received your examples of small objects.``
> >
> > We have already sent the link to ACs,  maybe they have not read the message. As the discussion period is going to end we will not be able to make further responses. Please remember to ask ACs for the results if they forgot to send you the link.
> >
> > By the way, we believe that the examples in the second row of the rebuttal PDF like "candles" and "socks" are already small. Considering that the model receives a "source mask" to indicate the to-edit region, the difficulties for the model would be similar even for more complicated scenarios.
> >
> > In addition, we appreciate your constructive discussions and we would make our paper clearer according to your suggestions.

---

### Official Review · Reviewer_8FSf · 2024-07-09

**Soundness:** 3
**Presentation:** 3
**Contribution:** 3
**Rating:** 7
**Confidence:** 5

**Summary:**

To achieve more precise imitative editing, the paper proposed a training framework, called MimicBrush. It randomly selects two frames from a video clip, masks some regions of one frame, and learn to recover the masked regions using the information from the other frame. Also, it constructs a benchmark, consisting two main tasks, i.e., part composition and texture transfer, to evaluate the performance comprehensively.

**Strengths:**

- The setting for part-level image editing is a novel idea. I believe it will be very helpful for designers to leverage imitative editing in their use cases.

- The paper provides comprehensive experiments to prove its effectiveness.

- The paper is well-written and easy-to-follow.

**Weaknesses:**

The definition of inter vs. inner is unclear.

**Questions:**

1. How is rank calculated in table 2?
2. Regarding line #195, does it mean that when the randomly selected grid number is 4, it would drop 4x75%=3 for SIFT-matched features and 4x50%=2 for other regions? So, in total, it would drop 5 grids?
3. What does the gid ratio mean in line #261?
4. The paper structure can be improved. Sec. 4.4 and Sec. 4.2 contain the “Qualitative comparison”. I suggest having only one Qualitative comparison section and separating it into subsections: “comparing with others” and “diverse applications without comparing with others”.
5. The mask representation is hard to identify, e.g., the right case in the first row or the left case in the third row in Fig. 7. I suggest to change a clearer representation.

**Limitations:**

The authors have shared the limitations and potential societal impact.

---

> ### Author Rebuttal · Authors · 2024-08-07
>
> Thank you for your acknowledgment and constructive suggestions. We will follow your recommendations to polish our paper for the revision.
>
> `` W1. The definition of inter vs. inner is unclear.``
>
> **Inter-instance** refers to cases where the source and reference images are of different instances, such as compositing the ear of a "cat" onto the head of a "man". This track corresponds to the most interesting and fantastic applications. However, since there is no ground truth for the composited target (e.g., the given man with the given cat's ear), we evaluate this track using metrics like DINO/CLIP scores (lines 174-176).
>
> **Inner-instance** refers to cases where the source and reference images are of the same instance. This track is relevant to post-refinement applications. For example, we might use AnyDoor or DreamBooth to generate a new image of a given bag and find that the generated logos or textures differ from the reference image. Our method can use the reference image to repaint the flawed region, even if they are not in the same pose. In our benchmark, we use real image pairs of the same instance as the source and reference images, manually masking some regions of the source image. In this scenario, the unmasked source image itself serves as the ground truth, allowing us to calculate SSIM, PSNR, and LPIPS.
>
> We will make it clearer in the revision.
>
>
>
> ``Q1. How is rank calculated in table 2?``
>
> We directly average the rank numbers for all test examples, which allows us to compare different methods effectively.
>
>
> ``Q2. Regarding line #195, does it mean that when the randomly selected grid number is 4, it would drop 4x75%=3 for SIFT-matched features and 4x50%=2 for other regions? So, in total, it would drop 5 grids?``
>
> As mentioned in L195, “we randomly choose the grid number N×N from 3 to 10”, thus if we select grid number is 4, we have 4 x 4 = 16 patches.  We first split them into SIFT-matched patches (e.g., 4 patches) and no-matched patches (e.g., 12 patches).  Then, we drop  4 * 0.75 = 3 matched ones and 12 * 0.5 = 6 non-matched ones. We will make it clearer, thank you for your suggestions.
>
> ``Q3. What does the gid ratio mean in line #261?``
>
> The grid ratio here refers to the ratios of masked grid patches, which correspond to the results in Tab.4. We report the results for masking 25%, 50%, and 75% of grid patches. We will clarify this in the text. Thank you for your suggestions.
>
> ``Q4 & Q5. The paper structure and mask representation.``
>
> Thank you for your suggestions. We will reorganize the paragraphs and use more effective ways to present the masks, such as using different colors or providing additional binary masks.

---

> > ### Comment · Reviewer_8FSf · 2024-08-12
> >
> > Thank the author for providing further information and contributing great work to the image editing community! I have read other reviewers' comments and the rebuttal information. I would like to remain my rating as 'Accept.’

---

> > > ### Author Response · Authors · 2024-08-12
> > >
> > > Thank you very much for your recognition of our work. We greatly value the feedback you provided, and we will follow the discussion to make this paper clearer and better.

---

### Official Review · Reviewer_KHnJ · 2024-07-10

**Soundness:** 3
**Presentation:** 3
**Contribution:** 3
**Rating:** 5
**Confidence:** 4

**Summary:**

The paper introduces a novel approach called imitative editing aimed at enhancing user creativity in image editing tasks. Traditional image editing often involves matching references to the source image, which can be challenging. In contrast, imitative editing allows users to directly draw inspiration from in-the-wild references without needing to precisely align them with the source image. This paper introduced a new approach called MimicBrush, a generative training framework that leverages video frame pairs to learn semantic correspondence and recover masked regions in images.

**Strengths:**

--This paper proposed a novel task, Imitative editing, which is a new editing scenario that enables users to leverage diverse reference images without the need for exact matches, promoting more intuitive and less restrictive editing interactions.
--The training strategy is self-supervised by utilizing two frames coming from the same video is intuitive. To ensure effective training, the paper further introduced data selection that select pairs without too big or too small variations, data augmentation to increase the task difficulty, and masking strategy to mask patches similar to reference in order to force the model to take guidance from reference for reconstruction.
--This paper proposed a test benchmark containing part composition and texture transfer to evaluate the performance of imitative editing, which is good.

**Weaknesses:**

--Regarding part composition and texture transfer, if I understand correctly, the proposed framework appears capable of simultaneously addressing both tasks. However, based on the training procedure, I am curious about how the model distinguishes between referencing parts and textures. This differentiation does not seem explicitly addressed in the training process, potentially leading to the model misinterpreting user intentions during the inference stage.

--Regarding ablation experiment, this paper present the training strategies about training data, augmentation and masking method. It would be better to also ablate the necessity of image training data. And for all the ablations, it would better to have qualitative comparisons to illustrate how such training strategy affect the visual quality.

**Questions:**

please see above weaknesses. Overall I think this paper is trying to address a new and interesting editing scenario, which will make image editing more convenient.

**Limitations:**

The authors have adequately addressed the limitations.

---

> ### Author Rebuttal · Authors · 2024-08-07
>
> Thank you for your acknowledgment and constructive suggestions, we will follow you suggestions to polish our paper for the revision.
>
> ``How the model distinguishes between referencing parts and textures.``
>
> During inference, users have the option to enable "depth control." The depth map serves as a strong condition for maintaining the shape of the original content. When enabled, the model preserves the original shape, thus forced to conduct texture transfer. Without the depth map, the model has no constraints to retain the original shape, allowing it to change the shape thus performing part composition. During training, we randomly drop the depth map to ensure the model is compatible with both tasks.
>
>
> ``Ablate the necessity of image training data``
>
> Thank you for your constructive suggestions. We have found that image data is also important for our methods. Please refer to **C. Importance of SAM Data** in the global rebuttal block for details.
>
>
> ``Qualitative comparisons for the ablation study``
>
> We agree that qualitative comparisons can help readers better understand our methods. In fact, we have already included qualitative ablations for different training strategies in Fig.A2 of the appendix. We will add more examples in the revision. We also add visualized ablation studies for the training data in the rebuttal PDF Fig.R3.

---

> ### Author Response · Authors · 2024-08-12
>
> Dear Reviewer,
>
> We kindly remind you about our submitted rebuttal. We understand that your time is valuable, and we greatly appreciate the effort you put into reviewing our work. If you have any questions or require further clarification on any points, please don't hesitate to reach out. We look forward to your feedback and hope for a favorable consideration.

---

> > ### Comment · Reviewer_KHnJ · 2024-08-13
> >
> > Thanks for the detailed response. The rebuttal addressed my concerns.

---

### Official Review · Reviewer_iASj · 2024-07-11

**Soundness:** 3
**Presentation:** 3
**Contribution:** 3
**Rating:** 4
**Confidence:** 5

**Summary:**

The paper proposes a new form of image editing, termed imitative editing. In this scenario, the user can edit the local area of the source image with a similar area to the reference image. This requires the system to automatically figure out what to expect from the reference to perform the editing. To achieve this goal, the author first designed a new generative network, dubbed MimicBrush. Then, the paper also constructs the benchmark from the video clip. The paper demonstrates both qualitative and quantitative results to validate the effectiveness.

**Strengths:**

1. The proposed imitative editing is reasonable, novel, and inspiring. As explained in the paper, such an editing form can facilitate other real-world applications in fashion and product design.


2. The paper contributes a valuable benchmark for the proposed task, including diverse topics and examples, which can benefit later research.


3. The demonstrated examples show desired and good qualitative and quantitative editing results.

**Weaknesses:**

# Motivation & Novelty
**Unclear motivation and advantages**: The core learning algorithm of the paper is an imitative U-Net, a reference U-Net, and a depth model. However, the advantages and motivations of such designs are not clearly discussed and validated. For example, some other networks such as Controlnet and IP-adapter, can provide conditional information from the reference image. Why is injecting features from the imitative U-Net better on the proposed imitative editing task?

**Unclear novelties in terms of learning compared with existing methods**: The proposed imitative task is new in that the system can edit part of the object whereas the latest methods mainly insert full objects. However, the novelty and difference seem to only lie in the training data (full object v.s. part object). The learning paradigm seems to be the same. Can these existing works do well in the proposed imitative editing task if trained with the collected part-level data? The essential difference **in terms of learning and formulation** between part-level editing and full-object editing is not addressed. My understanding is that they are essentially the same learning paradigm but in different concrete forms.


# Formulation & Writing
**Unclear notations**:
1. The paper proposes a new task and a new framework. However, the problem and algorithm are not rigorously formulated. For example, what are the input, output, and masks **mathematically**?

2. What is the training objective (loss function) in the pipeline? What is the loss used in the depth model?

3. For concatenation and injection of K, V from the reference U-Net, which layers are injected? Are cross-attention and self-attention processed in the same way?


# Experiments & Evidence
**Limited editing types**: The demonstrated editing types are limited.
1. For example, for the same source image, can the user edit different parts of the object with different reference images?
2. Does the proposed method support general image editing such as changing objects (cat to dog), or shape (round cake to square cake)?

**Unclear motivation on the SAM data**: The paper mentions using the SAM data in training except for the Video clips. How does the SAM data help the learning in this task? I think after the strong augmentation, the spatial position of different objects does not change. So, how does this help to learn the correspondence between the source image and reference image?

**Questions:**

Please refer to the weakness part.

My main concerns are about the depth of discussion and motivation of the proposed network and task, and the writing and formulation that are not serious enough to reach the criterion of Neurips.

**Limitations:**

The author already addresses the limitations of the work.

---

> ### Author Rebuttal · Authors · 2024-08-06
>
> **Motivation & Novelty**
>
> We disagree. There exist misunderstandings about our motivation and novelty. The "structure of dual U-Net" and the "data difference" are not our contributions. Please refer to **A. Core contributions and novelty** in the global rebuttal to see our motivation.
>
> ``1. Motivation for the dual U-Nets.``
>
> We explain the issues of Reference U-Net:
>
> - **First, Reference U-Net is a common practice, we do not claim contribution.** As mentioned in line 121, Reference U-Net is widely used in many tasks that require a reference image [56, 17, 46, 6, 58, 45]. These works have already proved the advantages of this structure over previous works like IP-adapter or controlnet. We use Reference U-Net solely as our feature injection module and do not claim it as our novelty, so we provide citations and do not focus on this part to discuss.
>
> - **Second, we have compared different strategies in Tab.3 and Fig.6.** "CLIP" represents the strategy of IP-Adapter, and "DINOv2" represents the strategy of AnyDoor. We observed that the Reference U-Net preserves details better, as it provides feature maps with higher resolution.
>
> ``2. Novelties in terms of learning.``
>
> The core difference is not the data, but the "mask-free" task formulation and the training strategy for "correspondence learning".
>
> Please refer to **B. Motivation of getting rid of the reference mask** in the global rebuttal block to see the differences between the previous object-level insertion methods.
>
> ---
>
> **Formulation & Writing**
>
> ``1. The input, output, and masks mathematically``
>
> Our task formulation is clearly demonstrated in Fig. 1, 2, 5, 6, 7. The input and outputs are straightforward: the model takes a source image $I_{source}^{h \times w \times 3}$, a binary source mask (indicating the region to edit) $M_{source}^{h \times w \times 1}$ , and a reference image $I_{ref}^{h \times w \times 3}$, and then predicts the edited image after iterative denoising.  The input and output could be formulated as follows:
>
> $  I_{result}^{h \times w \times 3} = f(I_{source}^{h \times w \times 3}, I_{ref}^{h \times w \times 3},  M_{source}^{h \times w \times 1} ) $
>
> ``2. What is the training objective in the pipeline? What is the loss used in the depth model?``
>
> We use the default loss function of diffusion models, which predicts the added noise at each step. The depth model is frozen during training (without loss). We did not emphasize these aspects as we did not make any modifications from the stable diffusion baseline.
>
> As suggested, we include some basic formulations of diffusion models.
>
> First, the ground truth target image( unmasked source image, with augmentation ) and the reference image, are encoded into latent space via VAE encoders. We note the target latent and reference latent as $\mathbf{z}^{tar}_0$, and $\mathbf{z}^{ref}_0$ at time step 0. During training, we randomly sample a timestep $t$ and add Gaussian noises $ \epsilon $ on the target latent:
>
> $ \mathbf{z}^{tar}_t = \sqrt{\bar{\alpha}_t} \mathbf{z}^{tar}_0 + \sqrt{1 - \bar{\alpha}_t} \boldsymbol{\epsilon}\ $
>
> We note the U-Nets as $ \boldsymbol{\epsilon}_{w} $, which takes the target latent(imitative U-Net), the reference latent(Reference U-Net), the CLIP image embedding of the reference image(noted as $c^{ref}$), and timestep $t$ as inputs to predict the added noise with MSE loss：
> $ \mathbf{L}_t = \left| \boldsymbol{\epsilon}_w(\mathbf{z}^{tar}_t, \mathbf{z}_0^{ref}, c^{ref}, t) - \epsilon \right|^2_2\  $
>
> We will include these details in the revised version.
>
>
> ``3. For concatenation and injection of K, V from the reference U-Net, which layers are injected? Are cross-attention and self-attention processed in the same way?``
>
> As discussed in "Motivation and Novelty," reference U-Net is a common practice [56, 17, 46, 6, 58, 45]. We do not claim contribution so we provide citations for the implementation details. Our implementation follows previous work[45] that injects K,V into self-attention layers of unet decoder layers. We will add these details in the revision.
>
> ---
>
> **Experiments & Evidence**
>
> ``1. For the same source image, can the user edit different parts of the object with different reference images?``
>
> Yes, there are two ways to achieve this:
>
> 1) concatenate multiple reference images like Fig.7 (the bottom-down example), where we concatenate the two reference images and mark multiple regions to be edited. The model can handle this case in a single pass.
>
> 2) edit different parts progressively in multiple turns, like Fig. A4 in the appendix (the first example of Hinton), we show how "cat ears," "scarf," and "glasses" are added from different reference images to the source image.
> **We also add more examples as required in Fig.R1 in the rebuttal PDF file.**
>
>
> ``2. Does the proposed method support general image editing such as changing objects (cat to dog), or shape (round cake to square cake)?``
>
> Image editing is a broad topic, and our goal is not to tackle all editing tasks. In this paper, we propose a novel form of imitative editing that encompasses "part composition," "texture transfer," and "post-refinement" for general categories. This already represents a wide scope, and no previous methods have achieved this combination.
>
> Examples for changing objects are added in Fig.R2(first row) in the rebuttal PDF file.
> Besides, although we could not support all the editing tasks, we could act as their post-refinement to enhance the performance. As in the last row of Fig.7, we could refine the artifacts of AnyDoor(which conducts object changing) and Cones-2 (which could change the shape and attributes of customized objects).  More examples of post-refinement are given in Fig.R2 in the rebuttal PDF. Our method could enhance the performance for a wide range of editing tasks only if they require a reference image.
>
> ``How does the SAM data help the learning in this task?``
>
> Please refer to **C. Importance of SAM Data** in the global rebuttal block.

---

> ### Author Response · Authors · 2024-08-12
>
> Dear Reviewer,
>
> We would like to gently remind you of our submitted rebuttal. We understand the demands on your time and sincerely appreciate the effort you invest in reviewing our work. If there are any questions or if you need further clarification on any points, please feel free to reach out. We look forward to your feedback and hope for a positive outcome.
>
> Thank you for your time and consideration.

---

> ### Author Response · Authors · 2024-08-13
>
> Dear Reviewer iASj,
>
> As the deadline for the discussion phase approaches, we wanted to kindly remind you to read our rebuttal. We highly value your feedback and are eager to incorporate your suggestions to improve our work.
>
> If there are any aspects that require further clarification or additional information, we would be more than happy to engage in continued discussion and provide any necessary details.

---

> > ### Comment · Reviewer_iASj · 2024-08-14
> > **Response to the rebuttal**
> >
> > Dear authors,
> >
> > I have read the rebuttal and other reviewer's comments. Thanks for the efforts in the experiments and clarification!
> >
> > My comments mainly proceed from the novelties and contributions in learning, i.e., theoretical understanding and derivation, OR the validated design of approaches and insights.
> >
> > Thus, the main components (Dual-Unet, KV replacement) are existing techniques (as also stated in RWq758). I acknowledged the authors' insights on "the Reference U-Net preserves details better, as it provides feature maps with higher resolution" in rebuttal. However, this argument is generally beneficial for all image tasks and does not sufficiently address how it facilitates the proposed task. So, the paper provides me with an impression of using existing (or sort of new) techniques and strategies with new data to build an application. Since this is not a theory paper, I think the novelties in the design of approaches can be limited though I acknowledged the interesting editing effects added in the rebuttal.
> >
> > Besides, considering the overall formulation of the current version of the paper, I choose to maintain my score.

---

> ### Author Response · Authors · 2024-08-14
>
> Dear Reviewer,
>
> Thank you for your response. However, we **disagree** with your comments, and **it seems that our rebuttal is ignored**. This comment does not involve the new information that we claimed in the rebutall.
>
> ``I acknowledged the authors' insights on "the Reference U-Net preserves details better, as it provides feature maps with higher resolution" in rebuttal. However, this argument is generally beneficial for all image tasks and does not sufficiently address how it facilitates the proposed task.``
>
> We have responded in the rebuttal that:
>
> "We have compared different strategies in Tab.3 and Fig.6. "CLIP" represents the strategy of IP-Adapter, and "DINOv2" represents the strategy of AnyDoor. We observed that the Reference U-Net preserves details better, as it provides feature maps with higher resolution."
>
> In this case, we conducted detailed ablation studies to explore the structure and gave both qualitative and quantitative analyses in the paper.  Considering this structure is **not our contribution**, we believe this experiment is already sufficient, and the advantage of this structure is exactly verified "in the proposed task".  However,
>  our claims around this experiment seem to be ignored,  so we disagree with this comment.
>
> `` So, the paper provides me with an impression of using existing (or sort of new) techniques and strategies with new data to build an application``
>
> We have repeatedly claimed in the rebuttal that **the dual U-Net is not our contribution or novelty** and carefully explained **the major differences are beyond the data**.  However, these explanations seem to be ignored, and the new comments still focus on **the novelty of the Reference U-Net and training data**. It seems that the misunderstanding still strongly exists.
>
> To make our contribution clearer, we would add a small paragraph at the end of the introduction section to summarize our contributions and novelties. In addition, we would add an explanation paragraph in Section 3.2 to claim the dual U-Net is not the contribution of this paper in the revised version.
>
>
> We appreciate your comments and please carefully read our responses and the initial rebuttal to see if these concerns have been addressed. We would be happy to get your suggestions to polish our paper. Thank you.

---

### Author Rebuttal · Authors · 2024-08-06

We thank all the reviewers for their valuable suggestions. We emphasize the motivation and novelty in the global rebuttal and  respond to each reviewer individually. Additionally, we include more figures and tables in the attached PDF file.

---

### A. Core contributions and novelty.

We would like to reaffirm our core contributions below.

- **Novel form for image editing.** Unlike the commonly studied task of "object insertion", our proposed "imitative editing" offers a new editing form, which does *not* require users to specify the region of interest from the reference image. Such a design **sufficiently simplifies the editing process**, because users only need to brush once (*i.e.*, the source image) instead of twice (*i.e.*, both the source image and the reference image), and hence clearly distinguishes our approach from existing methods. It is also noteworthy that getting rid of the reference mask is essential for *part-level editing*, making our algorithm more **flexible** in practice.
- **Novel training strategy.** To accomplish "imitative editing", it requires the model to **automatically** analyze the semantic correspondence between the to-edit region of the source image and the "full" reference image. For such a purpose, our approach benefits more from the training strategy but not the data. Concretely, we propose to leverage the temporal consistency and variation across video frames, and urge the model to recover the masked region of one video frame by *only* using the information from another frame *without* specifying the corresponding region. To our knowledge, our proposed training pipeline offers a brand new way of using video data for diffusion-based editing.
- **Benchmark for the novel task.** We construct a diversified benchmark that contains multiple tracks and covers a large variety of scenarios. Besides, we come up with various metrics to thoroughly evaluate the compare different methods. Our benchmark could be beneficial for further explorations for imitative editing.

---

### B. Motivation of getting rid of the reference mask.

As discussed above, this work presents a novel editing form, which allows users to *not* specify the region of interest from the reference image. Here, we would like to highlight **the necessity and the practical significance** of such a design.

- It is challenging for users to segment some local regions like tattoos or thin necklaces, even with intelligent tools.
- Local regions like tattoos are inherently intertwined with the context and difficult to understand when isolated.
- During training, methods like AnyDoor require image pairs with the same instance in two images with masks. While it's easy to collect object-level pairs through tracking or matching, gathering high-quality mask pairs for general local parts is of high cost. By removing the reference mask, we can independently mask one video frame and use another frame (without the corresponding mask) to train the model.

---

### C. Importance of SAM Data

SAM data contains the masks, we randomly choose a mask and then make two different crop (with different box sizes and locations) around the mask to make an image pair. Besides, we add strong augmentations to guarantee the variations. In this case, the masked regions are contained in both the source and reference images, but in different locations and sizes.

We find the SAM data important for the training procedure in the following aspects:

- Increased Diversity: Image data helps expand the diversity of categories and scenarios. It is challenging to collect category-balanced, high-quality videos at scale, but it is relatively easy to use open-source image datasets like SAM.
- Mask Compatibility: Image datasets like SAM contain large-scale masks. Using these real masks makes the model more compatible with user-drawn masks of arbitrary shapes.

**We add qualitative ablation results in the rebuttal PDF (Fig. R3) and quantitative ablations in the rebuttal PDF (Tab. R1).**  The results show that SAM data and video data are both important.

---

### Decision · Program_Chairs · 2024-09-25

**Decision:**

Accept (poster)

**Comment:**

The paper presents a new form of image editing called imitative editing. The review comments are mixed. Overall, the reviewers appreciate the formulation of the novel task and the proposed evaluation for this task. However, they have different opinions on the technical contributions. Specifically, the dual U-Net architecture and KV replacement are existing techniques. As the authors stated, architectural design is not the main contribution. The interesting bit about this work lies in the training strategy that uses sampled frames from a video and training the model without correspondence information. The authors' rebuttal did clarify these points and provided additional examples of small objects.

After the rebuttal, the reviewers' ratings remain diverged. The AC agrees that the paper's contribution is not in architecture design and data, but in the training strategy for this new task. As the new task is appropriately motivated and the evaluation is solid, the AC believes that this work would lead to more exploration and improvement on the new proposed image editing formulation. The AC thus recommends to accept.